# A Review of "Green Building" Regulations, Laws, and Standards in Latin America

**Carlos Zepeda-Gil** and **Sukumar Natarajan** *

Department of Architecture and Civil Engineering, University of Bath, Bath BA2 7BA, UK; cazg20@bath.ac.uk
* Correspondence: s.natarajan@bath.ac.uk

**Abstract:** Latin America covers 20% of the world's surface but only produces 12% of global carbon emissions. However, countries such as Brazil and Argentina have seen some of the most aggressive increases in individual country $CO_2$ emissions over the last two decades. Given that 80% of Latin America's population lives in cities, where we can expect the greatest increases in demand for energy and predicted growth in built floor space, it is necessary to ensure that these do not result in an overall growth in carbon emissions. Hence, we present the first review of the various "green building" rules developed in this region to curtail energy or carbon. This covers nine countries representing 80% of the region's population. We find that these countries in Latin America have developed 94 different green building rules, though to different extents. Many pertain to domestic buildings that are known to contribute 17% of the overall carbon emissions. Subsidies and tax incentives are most common, whereas innovative carbon market schemes have only been adopted in Mexico and Chile. In Argentina and Chile, regulations are similar to their European cold-climate counterparts but are poorly enforced. Overall, we find considerable progress in Latin America to create new standards and regulations, with enforcement being a major future challenge.

**Keywords:** green building regulations; building energy; Global South; Latin America

## 1. Introduction

Growing urbanisation and industrialisation have contributed to a global rise in carbon emissions. This is an unprecedented situation that has led to global warming becoming the most significant threat to humankind [1–3]. Buildings contribute a quarter of global carbon emissions [4]. These arise from energy demand during construction and operation, and hence, considerable effort has gone in the Global North to reduce energy consumption in buildings [5] as it is responsible for a third of the world's consumption. However, the Global South, which is expected to double global built floor space by 2050 [6], has only recently started to pay attention to this problem [7]. Unfortunately, for these countries, the presence, enforcement, and impact of green building regulations are documented, at best, poorly. In Latin America (LATAM), for example, buildings are thought to consume 22% of the total final energy demand of the region [8], but this is less well understood than in countries in the Global North. Estimates for the region suggest that energy demand will increase by at least 80% in 2040, compared to current demand [9], largely driven by an expansion of the middle class [10].

However, shared languages, history, and culture with relatively open borders, especially in South America, suggest the possibility of curtailing this trend through shared information and expertise. As major emitters of carbon, buildings could play a key role in mitigating the effects of climate change [11] from the construction stage [12], throughout their "useful life" [13], and to the end of their lifecycle [14]. These stages must be looked at in detail when addressed as their boundaries are ambiguous and may raise more problems if they are not looked at in detail [15]. Green building rules (GBRs) focus on one or more of these attack points across the lifecycle, but the overall picture of current

efforts being made in different countries is missing. There is, therefore, an urgent need to undertake a review of the green building rules in this region to enable better planning for climate change mitigation and adaptation.

Hence, the aim of this paper is to provide a systematic review of the current GBRs in Argentina (ARG), Brazil (BRA), Chile (CHI), Colombia (COL), Costa Rica (CSC), Guatemala (GUA), Mexico (MEX), Panama (PAN), and Peru (PER). Together, they represent 80% of the region's population, 84% of the total surface area, and 87% of the region's GDP. This enables us to create a comprehensive picture of the current state of play in the region.

We define a green building rule to be any official governmental rule aimed at reducing the energy or carbon performance of a building or improving its broader sustainability. This is to allow the review to capture a wide variety of initiatives aimed at positively affecting current or future building performance in the region. A common approach to deal with GBRs is to focus on a particular aspect, such as energy rating systems (ERSs) [16], or renewable energy and energy management [17]. These are included within the scope of our review; however, ours is broader as it includes other aspects such as technical standards, social housing policies, or national energy-reduction policies.

GBRs have been adopted worldwide as a strategy to reduce energy consumption in buildings. There are many aspects that determine their success such as the country's economic situation, public health benefits, and political acceptability [18]. For instance, a crisis-stricken developed economy, such as Spain, enacted several "Royal Decrees" to promote the use of renewables in buildings, but their economic situation led to a decrease in its usage [19]. In contrast, a newly developed country, such as China, implemented national policies that led to an increment of their use of renewables in buildings [20]. GBRs have been implemented in other parts of Asia but without much success. Taiwan passed the "Frameworks of Sustainable Energy Policy" that includes similar strategies to the ones adopted in LATAM but without much success yet [21]. In some places of Africa, the policies towards energy reduction in "low-income" buildings were very well accepted, in particular biofuel. Mandelli [22] explains that this was because people could relate it to traditional fuel types (i.e., firewood, dung, charcoal). This experience should be used in regions of LATAM where the use of traditional fuel types is still a common practice.

## 1.1. Global Context

LATAM covers a fifth of the total world's surface; however, it only consumes 6% of the world's total energy at present (Table 1). All the countries in the region are developing economies, and energy demand is known to rise with economic growth [23–26]. This is not accidental, as the rate of increase in energy use is often seen as a direct metric of a growing economy in many developing countries [27].

**Table 1.** World's relative primary energy consumption by region in 2018. All columns, except the last, are normalised to the North American total (set as 1) (source: World Bank [23]). O = oil, G = natural gas, C = coal, N = nuclear, H = hydroelectricity, R = renewable, % = percent of world total (e.g., Latin America = 0.336/5.242 = 6%).

| Region | O | G | C | N | H | R | Total | % |
|---|---|---|---|---|---|---|---|---|
| North America (US + Canada) | 0.389 | 0.303 | 0.125 | 0.081 | 0.058 | 0.043 | 1.000 | 19% |
| LATAM | 0.151 | 0.084 | 0.018 | 0.003 | 0.065 | 0.015 | 0.336 | 6% |
| Europe | 0.281 | 0.178 | 0.116 | 0.080 | 0.055 | 0.065 | 0.775 | 15% |
| Eurasia | 0.073 | 0.189 | 0.051 | 0.018 | 0.021 | 0.000 | 0.352 | 7% |
| Middle East | 0.156 | 0.180 | 0.003 | 0.001 | 0.001 | 0.001 | 0.341 | 7% |
| Africa | 0.072 | 0.049 | 0.038 | 0.001 | 0.011 | 0.003 | 0.174 | 3% |
| Asia Pacific | 0.641 | 0.268 | 1.074 | 0.047 | 0.147 | 0.085 | 2.263 | 43% |
| World | 1.763 | 1.251 | 1.426 | 0.231 | 0.359 | 0.212 | 5.242 | - |

Such an approach not only conflates "consumption of" with "access to" energy, which is arguably the key goal, but also does not discriminate between energy sources. The consequence of such an approach can be seen in Figure 1, which shows that newly industrialised economies such as China and Saudi Arabia

have nearly quadrupled their $CO_2$ emissions in just over two decades. Although the three Latin American countries in this group—Brazil, Argentina, and Mexico—have not seen such aggressive increases, they still experienced an average increase of 76%, 48%, and 30%, respectively, over the same period. While it is presently hard to disaggregate these data, there is little doubt that the total increase in $CO_2$ is a direct consequence of total energy consumption, a substantial proportion of which comes from buildings.

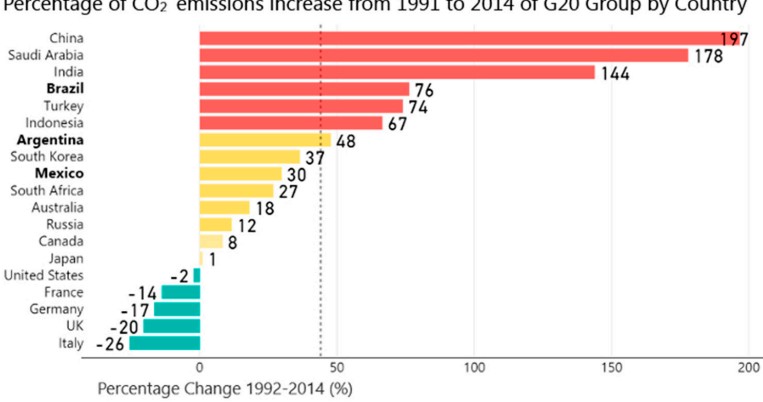

**Figure 1.** Percentage change in $CO_2$ emissions per capita within the G20 group from 1992 to 2014. We observe a dramatic increase in the newly industrialised (red), a moderate increase in emerging (yellow), and a decrease in the older industrialised economies (green). Countries from Latin America (LATAM) are in bold. Source: World Bank, 2016.

### 1.2. Energy Consumption in Latin America

LATAM is the world's most urbanised region with 80% of its population living in cities [28] and is projected to increase to 90% by 2050 [29]. Five of the ten largest urban areas in LATAM are located within just two countries: Mexico (Mexico City and Guadalajara) and Brazil (São Paulo, Rio de Janeiro, and Belo Horizonte). Indeed, these two countries consume 54% of the region's energy (Figure 2). The figure also demonstrates that most of the energy supply in LATAM is non-renewable. Hydroelectricity is a prominent source of energy in many countries—e.g., it is the second-largest source in Brazil—but is not usually considered a "green" source of energy owing to the negative environmental and societal impacts associated with it.

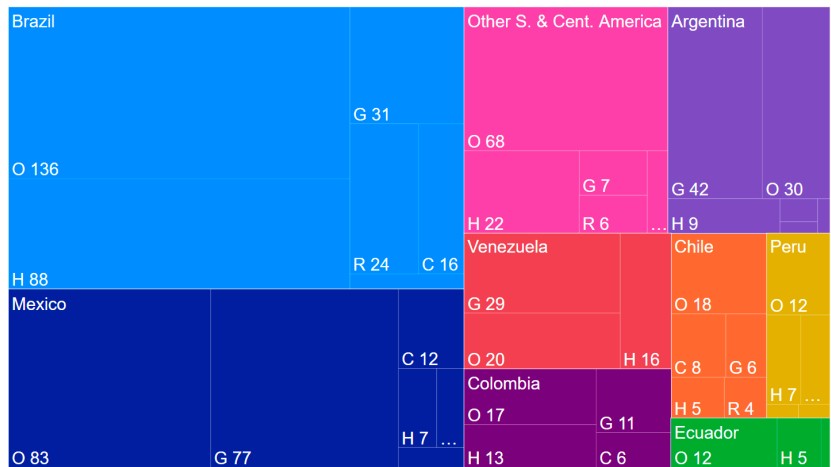

**Figure 2.** Energy consumption by country in Latin America in million tonnes of oil equivalent. One tonne of oil equivalent is the amount of energy released when one tonne of crude oil is burned and is equal to 11.6 Mwh, showing that Brazil and Mexico are the main energy consumers of the region. ("Other South and Central America" includes the remaining countries in Latin America). O = oil, G = natural gas, C = coal, N = nuclear, H = hydroelectricity, R = renewable. Source: World Bank, 2016 [23].

*1.3. Paper Structure*

The rest of this paper is divided into four sections. The next section, Methods, discusses a classification framework that can be used to analyse green building rules and our survey method. The results section describes the result of our survey of such rules in LATAM and the outcomes from applying the classification framework from the Methods section. Finally, we discuss the degree of success enjoyed by GBRs in LATAM and conclude by presenting the key lessons learned through this review and towards future work.

## 2. Methods

As the scope of our review is broad, GBRs can be expected to be highly diverse ranging from national-scale programmes to highly local policies. Hence, methods for classification and metrics for analysis are needed, described further below.

*2.1. Classifying Green Building Rules*

As stated earlier, we use the umbrella term "rules" to capture the wide variety of efforts to improve the energy performance, reduce carbon emissions, or improve the overall sustainability of buildings. There is considerable debate in the policy literature as to the best means of evaluating such rules [30]. Lowi's typology theory is a long-standing framework that has been used with varying degrees of success both for classifying and evaluating real-world policies [31]. While it continues to attract criticism—primarily due to failures in capturing real-world policies that do not fall easily into the categories suggested by the theory—it continues to inform practice amongst academics for policy analysis [32,33]. Hence, we use it as a useful starting point for our own analysis.

Lowi's basic premise is that any public rule is essentially a means of *coercion*, i.e., ensuring citizens or other actors behave in a manner envisaged by the rule maker. Hence, Lowi's typology theory is constructed over two axes: the *likelihood of coercion* ranging from the immediate to the future (vertical axis in Figure 3) and whether *coercion works through* individual conduct or the environment of conduct (horizontal axis in Figure 3). This results in four quadrants [34] representing:

- Regulatory policies: where rules *impose* obligations, with transgressions being deemed criminal (e.g., public health laws, industrial safety).
- Redistributive policies: where rules *impose* classifications/status/categories (e.g., income tax, national insurance schemes).
- Distributive policies: where rules *confer* unconditional facilities or privileges (e.g., public works, land grants, subsidies).
- Constituent policies: where rules *confer* power or authority (e.g., rule-setting governmental organisations, agencies for budgetary policy).

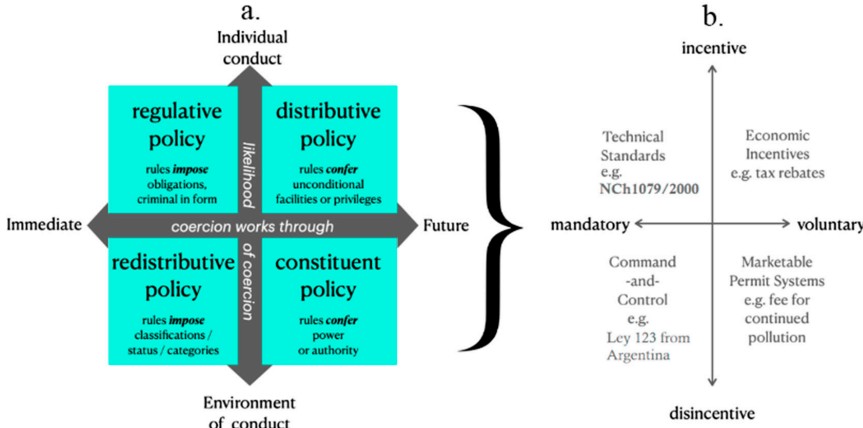

**Figure 3.** Lowi's framework (**a**) and our own framework (**b**) to classify and evaluate green building rules.

A pilot survey of the laws and decrees on green buildings in the selected countries suggested few rules neatly fit into the four quadrants of Lowi's framework, a well-known criticism, as above. For example, Brazil taxes property purchases at 4%, but this is reduced to 2% if the Qualiverde (Green Quality) certification [35] is obtained. On the one hand, the tax is a redistributive policy whereas the main action is probably one of a distributive policy (i.e., a subsidy or discount on the tax). A key issue with Lowi's framework is the use of time as a dimension (horizontal axis in Figure 3). Many of the rules identified in our pilot survey were not suggestive of a time dimension. Hence, we simplify Lowi's framework into two axes comprising of the *nature of incentivisation* versus *voluntary or mandatory* rules:

- Mandatory rules with disincentives: These can be thought of as command-and-control (CAC) rules where fines or other "punishments" are levied for rule-transgression. For example, the result of not complying with the Chilean Law "Ley No. 458" will result in a fine of 5% to 20% of the total cost of the project.
- Mandatory rules with incentives: These are usually technical standards (TS); for example, the Mexican Green Mortgage scheme. It is a mandatory low-interest loan added on top of a loan aimed to purchase green technologies.
- Voluntary rules with incentives: These are usually voluntary economic incentives (EIs) to adopt better rules. The Brazilian rule discussed earlier neatly falls into this category.
- Voluntary rules with disincentives: These are marketable permit systems (MPSs), i.e., rules that allow business-as-usual practices to continue, but at a cost; for example, Colombia's National Programme of Greenhouse Gas Tradable Emission Quotas (PNTCE), which allows a pollution allowance in exchange for a price set by auction [36].

To enable readability, the rest of the paper will refer directly to each quadrant of our framework using one of the four acronyms defined above (CAC, EI, MPS, TS).

It is noteworthy that several policies in the region are driven by nationally appropriate mitigation actions (NAMAs) [37]. NAMA projects, specifically directed at developing countries, were negotiated at the United Nations Framework Convention on Climate Change's (UNFCCC) Conference of Parties in Bali [38]. They can be subdivided into three types: unilateral (financed with a country's own resources. Targets should go in accordance with international goals, but they are not required to report results), supported (receive international financial and technological support. Should report their mitigation contributions through the "Biennial Update Report (BUR)"), and financed/credited (receive international support, but these are supported through the carbon market) [39]. NAMA projects have been adopted by Brazil, Chile, Colombia, Guatemala, and Mexico in LATAM. However, many will naturally fall into one of the above four categories due to the manner of implementation. For example, Mexico created "NAMA housing" as a low-energy residential building certification tool in collaboration with the Passivhaus Institute in Germany, an example of a TS. However, NAMA projects often go beyond green-building-related projects. For this paper, we only considered those that directly impact on building energy reduction and are described as a subsection of the type of policy according to our framework (e.g., EI-NAMA).

## 2.2. Survey

We undertook a survey of all GBRs in the selected countries through systematic searches of the academic literature in Spanish, Portuguese, and English, followed by accessing the original rules and policies from the mandating institution. This was enabled in significant part through the public availability of the rules on the internet and most of the technical standards (with the exception of the Argentinian IRAM Standards [40]). All green building rules enacted and promulgated to March 2020 were selected.

## 3. Results

Our survey resulted in 94 individual GBRs within the selected countries as shown in Table 2. The full list can be seen in Appendix A. Each of the 94 rules broadly emanate from national-level laws (i.e., constituent policy in Lowi's framework), as shown in Appendix B. Table 2 and Figure 4 show in detail the number of rules enacted by each country and its corresponding classification. This list is summarized according to our GBR classification framework in Appendix C.

**Table 2.** Total policies found by type and country. Countries are represented by three-letter equivalents per the list in Section 1.2.

| Type of Policy | ARG | BRA | CHI | COL | CRC | GUA | MEX | PAN | PER | Total |
|----------------|-----|-----|-----|-----|-----|-----|-----|-----|-----|-------|
| TS | 8 | 5 | 7 | 5 | 5 | | 11 | 3 | 4 | 35 |
| CAC | 6 | 3 | 4 | 1 | 2 | 1 | 1 | 1 | 6 | 25 |
| EI | 1 | 4 | - | 6 | - | 1 | 5 | 1 | 5 | 22 |
| EI-NAMA | - | 1 | 1 | - | - | 1 | 2 | - | - | 5 |
| MPS | 1 | - | - | 1 | - | - | 2 | - | - | 4 |
| Total | **16** | **13** | **12** | **13** | **7** | **3** | **21** | **5** | **15** | **94** |

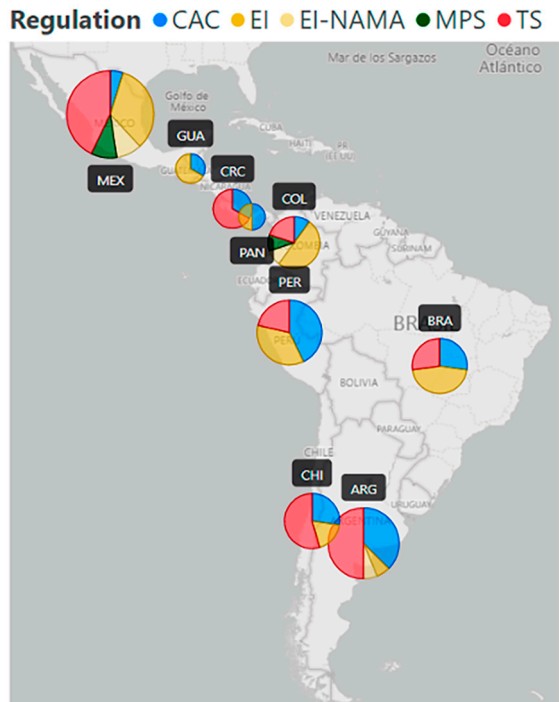

**Figure 4.** Geographical distribution of the green building rule (GBR) rule types in our review. The size of each bubble is proportional to the number of GBRs enacted to date and the colours show the distribution of individual policy types in each country.

### 3.1. Broad Chronology

Figure 5 shows a timeline of the policies enacted in the region, with numbers indicating the number of relevant policies by a given country in a single year. The first GBRs were enacted during the late 1990s and early 2000s. These include Argentina's IRAM 11.605 [40], Mexico's NOM-009-ENER-1995 [41], Chile's Ley No. 458 [42], the Brazilian Lei 10.295 [43], and the Colombian Standard NTC 5316 [44] (2004—simply a translation of the ASHRAE 55) and their Sello Ambiental Colombiano (Colombian environmental seal) [45]. None of these are considered to be successful due to, for example,

lack of expertise for their enforcement, strict regulations/standards, being economically unviable/with unattractive incentives, and corruption [46–49].

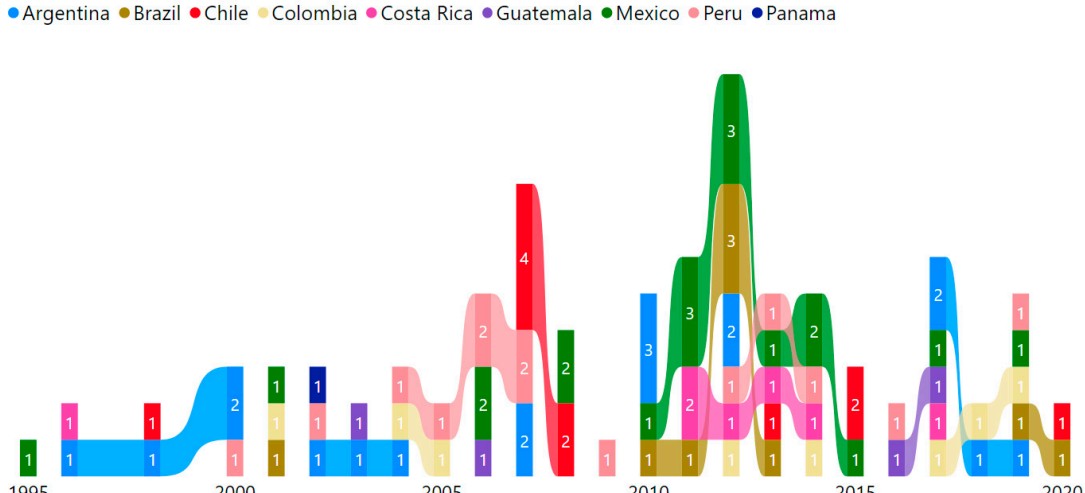

**Figure 5.** Chronology of the green building rules launched. Each number indicates the GBRs promulgated in a given country (coded by colour) in a given year.

The majority of the reviewed GBRs (36%) were enacted between 2010 and 2015 (as seen in Figure 5). Positive effects after Mexico's mandatory standardisation norms (NOMs), as well as with their "green mortgage" scheme, have also been reported [50]. These are now seen as being the first to produce measurable energy reductions. For example, the Mexican TS NOM-028-ENER-2010, which regulates energy efficiency in lamps, resulted in 11,782 GWh savings in 2013 [51], equivalent to the electricity use of 1.4 M (i.e., 4%) of homes for one year [52]. Similarly, its code for sustainable housing required a USD 130 M investment but produced a cumulative reduction of 13.3 Mt $CO_2$ from 2015 to 2020 [53], equivalent to the energy usage during one year of 1.5 M (4%) of houses [52]. In contrast, there were still some policies from that period that were not enforced or had negative outcomes due to various reasons. The Brazilian local law "Lei Municipal (Local Law) 6.793/2010", which granted fiscal incentives to those using green technologies, was heavily criticised due to limited access to green technologies in the Brazilian market [54]. Brazilian regulations, Law No. 8.666 (2011) and Law No. 3.5745/2012 (2012), that promote sustainable construction are said to be unviable, too strict, as well as prone to corruption [55,56].

Mexico's Federal Law for Climate Change (2012) delegated the responsibility to mitigate climate change to different governmental institutions according to their sectoral responsibility. The institution responsible for assuring buildings would reduce their energy consumption is the Ministry of Urban and Rural Development (SEDATU), which created the National Housing Commission (CONAVI) for this task. CONAVI created the "Hipoteca Verde" (Green Mortgage) and the "Esta es tu casa" (This is your house) programmes that provide additional finance to cover for green technologies and energy-efficient appliances [57]. The main weakness of this policy is that it does not establish any follow-up mechanisms to ensure its appropriate implementation [58]. This is known to be a major factor in the success of some standards such as the aforementioned Passivhaus (Passive House) standard, which includes rules for checking implementation [59]. Chile launched the National Programme of Energy Efficiency (PPEE) through National Decree No. 336 in 2006 and expects to save up to 110 GWh per year with a USD 75 B investment [60], enough to provide energy use for 9000 homes for one year [52].

Since 2015, the number of new GBRs have fallen. The overall trend is one of consolidating existing laws and encouraging uptake, though this may be at the expense of locking in poor efficiency. For example, Brazil renewed Decree No. 9.864, which establishes energy rating systems in buildings in

2019, making it less strict by requiring little insulation on roofs and none on walls. The Argentinian TS "IRAM 11900" (2017) established a simplified methodology to calculate energy efficiency in buildings and an energy labelling system. At present, it has only been launched on pilot tests in Rosario City (temperate and humid climate, Cfa-Köppen) with 500 homes and in San Carlos de Bariloche (cold and humid climate, Csb-Köppen) with 200 homes, with the objective to validate calculation methods and implementation [61]. The Guatemalan NAMA project "Efficient Use of Fuel and Alternative Fuels in Indigenous and Rural Communities" was launched with an investment of USD 15.1 M as a response to the high usage of firewood inside buildings (57% in 2010), known to increase the risk of respiratory and cardiac diseases. The aim of this project is to replace these stoves with energy-efficient appliances. As its performance seems to be positive so far [62], there are presently negotiations to allocate an extra USD 5.95 M [63].

*3.2. Energy Rating Systems*

An energy rating system (ERS) is a tool that allows the user to know the energy consumption of an appliance, as well as its level of energy efficiency. They range from "most efficient" (i.e., "A++" in green) to "least efficient" (i.e., "E" in red). The higher the efficiency level, the lower the energy consumption. This increases competition and permits buyers to identify and purchase the most energy-efficient products, allowing their faster integration into the market [64]. There are a growing number of similar studies that benchmark GBRs and, in particular, energy rating systems [65,66]. This proves not only the relevance of the topic but also the effectiveness of these when dealing with energy reduction in buildings.

Over time, market transformation occurs, with less efficient appliances seeing slower uptake. For instance, different ERSs were introduced in Mexico in 1995, and by 2005, these are estimated to have caused a decrease of 9.6% of total national energy demand compared to projected "business as usual" [67]. However, these gains have eroded over time through continued increases in overall demand resulting in a near-linear trend of an 11% increase in energy demand per annum between 1990 and 2018 [68]. Shared languages and cultures have allowed some countries to take inspiration from others in the region, suggesting the possibility of further "cross-fertilisation" in the future. For instance, Costa Rica and Panama's ERSs are aligned with the Mexican systems. However, the rest of the studied countries aligned with either the European Union (EU) or US rules, which are often seen as class leading [69].

ERSs were introduced in LATAM with the exception of MEX and COL (mid-1990s) during the early 2000s (and fully adopted by 2017), a decade later than the United States (US) and the European Union (EU). The late introduction of ERSs in the region impacted their development and evolution, as we found notable differences between these two. In LATAM, after the evaluation is completed, the evaluator provides the label to be publicly displayed, but the process finishes there. In contrast, on top of the label, the EU and US ERSs provide recommendations for building enhancements related to energy savings, along with its corresponding budget. These also provide a cost–benefit analysis with an amortisation table that includes payback [70]. Furthermore, ERSs in LATAM (with the exception of ARG) mainly focus on building performance, whereas those in the US and EU also look at their energy consumption. This may be because, in LATAM, there are still very few cases where the building has the necessary infrastructure to determine in detail the sources of energy consumed within the building. Another difference is that none of the evaluations in LATAM inform the $CO_2$ emission savings, unlike in the EU and US.

In terms of presentation, labels from ERS are traditionally divided into "comparative" and "scalar" [71]. The comparative style (see 'a' in Figure 6 used by ARG, BRA, and CHI) shows the energy performance of the evaluated item on a category (usually a letter or number) and is generally more "user-friendly". The scalar (see 'b' in Figure 6, used by MEX), includes information about the energy consumption, and its operative cost of the evaluated item on a continuous scale. Colombia uses a mix

of both showing information about the energy consumption and percentage of energy savings, as well as the usual rating system.

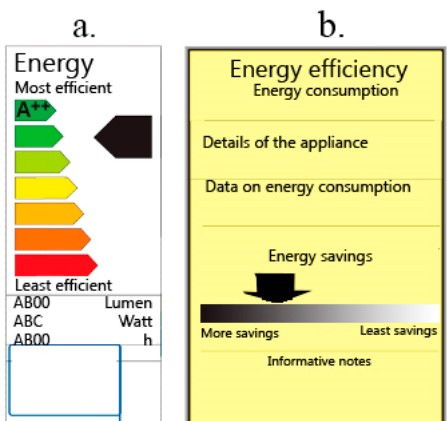

**Figure 6.** Two types of energy labelling stamps used in the reviewed countries, showing the comparative (**a**), and the scalar (**b**) types.

In LATAM, 60% of the end-use of electricity in buildings goes to lighting and keeping adequate internal temperatures; i.e., the usage of air-conditioning systems and the efficacy with which the building can maintain its internal temperatures [72]. At the moment, there are no energy rating systems that focus on heating or energy production. In some places like Argentina and Chile, these are included within the building's labelling system. We, therefore, only reviewed the ERS related to buildings, lighting, and air-conditioning systems. The themes used when benchmarking the different ERSs in the region are based on scope and weighting. Other similar studies that focused on ERSs had taken similar approaches. For instance, Mattoni [16] compared the main energy rating systems based on their scope and outcomes, given that each has their own calculation methods, point system, and weighting system. At the time of writing, Argentina and Chile have rating systems for building fabric, lighting, and air-conditioning systems. Brazil and Mexico do not have ERS for lighting systems. Costa Rica and Panama are currently developing their own, and Guatemala does not have any [72,73].

### 3.2.1. Building Fabric

Table 3 lists country-wise rules and the resultant fabric standards in LATAM. We find that most Latin American countries do have thermal performance regulations, apart from Panama (which is currently developing TS for building fabric) and Guatemala. However, these only aim to control fabric losses, ignoring heat losses by air leakage/ventilation. This means that the potential effects of any improvement in the thermal regulation can be affected by unintended consequences such as the creation of thermal bridging, reduced indoor air quality, excessively leaky envelopes, and comfort take-back [74]. As this is a region with a wide range of climates, from rainforests in the amazon or arid-desert-hot in the Sonora desert in Mexico, to polar cold in the Argentinian Patagonia, each country adapts specific technical requirements for buildings according to the climatic zones contained within them For simplicity, we present the nationally averaged U-values for the reviewed countries in Table 3, and the full list can be seen in Table 3. Note that all source R-values ($m^2K/W$) have been converted to U-values ($U = \frac{1}{R}$) for consistency.

**Table 3.** Comparison of the different rules that deal with heat loss in the building fabric, where M stands for mandatory (e.g., a minimum energy performance standard (MEPS)) and V stands for voluntary (e.g., an energy rating system (ERS)). Numbers in brackets for U-values indicate standard deviations and demonstrate climatic variations within the national average.

| Country Code | Rules | Scope | Building Design and Thermal Comfort | | | | | |
| --- | --- | --- | --- | --- | --- | --- | --- | --- |
| | | | National Average U-Values (Wm$^{-2}$K$^{-1}$) | | | | Comfort Model | Source |
| | | | Wall | Window | Roof | Floor | - | - |
| ARG | V—IRAM 11605 | TS on building insulation. Applicable for residential buildings at the national level | 1.03 (0.68) | -[1] | 0.66 (0.32) | 0.65 (0.61) | -[2] | [75,76] |
| | M—IRAM 11604 | TS on building insulation. Applicable to all buildings only in Buenos Aires | 1.03 (0.68) | - | 0.66 (0.32) | 0.65 (0.61) | - | [77] |
| BRA | V—NBR 15575 for residential buildings | Labelling in public buildings in 2020 and commercial in 2025 | 3.1 (0.84) | - | 1.3 (0.53) | - | - | [78] |
| | M—Decree: No. 18/2012-RTQ-R for energy labelling | TSs on heat loss, ventilation, and lighting in residential buildings | 2.93 (0.70) | - | 2.1 (0.17) | 2.1 (0.17) | -[2] | [79–81] |
| CHI | V—CES (certification of sustainable building) | Evaluation of the energy efficiency of public buildings in Chile | 1.93 (0.31) | - | 0.83 (0.10) | 0.83 (0.10) | -[2] | [82] |
| | M—Official Chilean Standard, NCh 888. Of 2001 | Establishes the calculation requirements for thermal performance of windows | - | 1.4 | - | - | - | [83–85] |
| COL | V—NTC ISO 50001 Colombian Technical Standard | Establishes minimum guidelines at the level of comfort, energy efficiency, protection of environment, and safety | - | - | - | - | Adaptive | [86] |
| MEX | V—NMX [1]-C-460-ONNCCE-2009 and NMX-C-7730-ONNCCE-2018 | NMX [3] standards for heat loss. Voluntary to all types of buildings | 0.55 (0.84) | - | 0.90 (0.27) | - | - | [87–89] |
| | M—NOM-020-ENER-2011 in residential, NOM-008-ENER-2001 in nonresidential | NOMs standards for heat loss. Mandatory to residential, commercial, and public buildings | 0.69 (0.15) | -[4] | 0.69[5] (0.15) | -[6] | PMV[7] | [90] |
| PER | V—Board Agreement No. 02-12D-2015-SUSTAINABLE HOUSING BONUS (BMS), and the supreme decree No. 026-2010-EM (2010) and Law No. 27345 | BMS is a certification tool for residential dwellings. The national decree is set to provide guidelines for energy reduction in dwellings | 2.27 (0.91) | - | 1.85 (0.64) | 2.78 (0.29) | - | [91,92] |
| CRC | - | - | - | - | - | - | - | - |
| GUA | - | - | - | - | - | - | - | - |
| PAN | - | - | - | - | - | - | - | - |

[1] The IRAM 11604 establishes a labelling system for energy efficiency in windows ranging from A (most efficient) to G (least efficient). However, it does not specify a particular U-value or K-value. Further information is in [77]. [2] Although we found several thermal comfort studies in these countries using both models (adaptive and steady state), we could not find an official rule that officialised a thermal comfort method in that particular country. [3] NMX stands for 'Mexican Standard (Norma Mexicana)', and it has a voluntary nature. [4] The standard [90] provides a calculation method for heat losses but does not specify a minimum value. [5] This is the average for buildings up to 3 floors; walls and roof are the same. [6] Floors in direct contact with land are not listed in the standard [90], nor included in the calculations. [7] PMV stands for Predicted Mean Vote, and it is one of the most accepted thermal comfort models internationally.

### 3.2.2. Natural Lighting

None of the reviewed countries have mandatory standards that specifically relate to minimum requirements of daylighting, nor do they consider the well-accepted daylight factor. Mexican NOM-025-STPS-2008 [93], Chilean Supreme Decree 594 [94], and Colombian Resolution No. 180540 [95] establish minimum levels of illumination levels in lux (lx). It is written on these standards that the lx thresholds must be met either with natural or artificial lighting, a similar case with the Argentinian standard IRAM-AADL J 20-06 [96] However, the IRAM standard does not allow spaces to be purely naturally illuminated; rather, they should be mixed.

The fact that to date there are no established daylight factors in the countries reviewed suggests that this issue is highly undervalued in the region. It certainly deserves more attention due to the positive impacts it has on the building and its occupants. On the one hand, it serves as a passive energy-saving strategy, reducing energy consumption from nonrenewables. On the other hand, it brings plenty of positive impacts on human health (healthy circadian rhythms, improves productivity, and reduces stress).

The International New Construction Guidelines of BREEAM (Building Research Establishment Environmental Assessment Method) [97] considers different average daylight factor (ADF) values depending on the latitude. The latitude of our studied region ranges from 32° in Tijuana, to −51° in the Argentinian Patagonia. However, on average, they suggest an ADF of 1.4% (SD of 0.16) in at least 80% of the building. Nonetheless, daylight studies in countries located in the tropics [98,99] (from mid-Mexico to North Argentina–Mid-Chile) suggest that buildings in these regions should be designed more in accordance with a larger daylight and glare control.

### 3.2.3. Artificial Lighting

For lighting, the trend towards the future is to continue developing TSs and ERSs for newer technologies such as LED lighting. Although some countries lack these, most of them include minimum energy performance standards (MEPS). Due to the high efficiency of LED lighting, it is expected for all the reviewed countries to develop their own TSs and ERSs. Table 4 includes the minimum energy efficiency values for both MEPS and ERS.

### 3.2.4. Natural Ventilation

The most internationally accepted standard for natural ventilation was EN 13779:2007. It has recently been withdrawn, and the part that deals with natural ventilation has now been replaced by the ISO 17772-1:2017 [100]. It specifies that the minimum airflow rate must not be less than 4 l/s/person [100]. In addition, the BREEAM New Construction Guide [97] specifies that to obtain the credit on internal comfort, the building must provide fresh air into the building. In naturally ventilated buildings, windows are 10 m away from external pollution sources. It also proposes that building ventilation strategies must be flexible and adaptable to meet the needs of the occupant. This is demonstrable by:

- The area of the openings, which must correspond to 5% of the internal area of the room.
- Cross-ventilation, and whether it is proven through the design

**Table 4.** Comparison of lighting standards in the region. We show the status (S) of whether a minimum energy performance standard exists (Y) and/or whether this is accompanied by labelling (L). Fluo stands for fluorescent lamps, CFL stands for compact fluorescent lamps, and HPS stands for high-pressure sodium. Source [60]. We also show the minimum energy efficiency values for the reviewed countries are in lumen per watts (lm/W) unless otherwise stated.

| Country | Type | Incandescent | | Halogen | | Ballast — Fluo | | Ballast — HPS | | Tube Fluo | | CFL | | HPS | | LED | |
|---|---|---|---|---|---|---|---|---|---|---|---|---|---|---|---|---|---|
| | | S | MEEV | S | MEEV | S | MEEV | S | MEEV | S | MEEV | S | MEEV | S | MEEV | S | MEEV |
| ARG | V | - | NK [8] | - | - | Y | NK [8] | - | - | - | NK [8] | - | NK [8] | - | - | - | - |
| | M | L | | - | | - | | - | | L | | L | | - | | - | |
| BRA | V | - | 15 | Y | 40 | Y | 50 | - | 20 | Y | 40 | - | 50 | - | - | Y | 96 |
| | M | Y, L | | - | | - | | Y, L | | - | | Y, L | | L | | - | |
| CHI | V | - | - | - | 8–17 | Y | - | - | 11–21 | - | - | - | - | - | - | - | - |
| | M | - | | - | | - | | - | | - | | - | | - | | - | |
| COL | M | Y, L | 12–15 | Y | 45 | Y, L | 45 | - | - | Y, L | 85 | Y, L | 55 | Y | 55 | - | - |
| MEX | V | Y | 20.69 | - | 60 | - | 86 | - | 75 | - | 86 | - | 20.69 | Y | 60 | - | 40 |
| | M | - | | Y | | Y | | - | | Y | | Y | | - | | Y | |
| PER | V | - | EI <60% [9] | - | - | Y, L | EI <60% | Y | - | Y | EI <60% | Y | EI <60% | - | - | - | IEE ≤0.13 |
| CRC | V | Y, L | 52 | - | - | - | 52 | - | - | - | - | Y, L | 52 | - | - | Y | - |
| | M | - | | - | | Y | | - | | - | | - | | - | | - | |
| GUA | - | - | - | - | - | - | - | - | - | - | - | - | - | - | - | - | - |
| PAN | - | - | - | - | - | - | - | - | - | - | - | - | - | - | - | - | - |

[8] The Argentinian IRAM standard does not provide open access to its standards; therefore, we could not access this information. NK stands for "not known". [9] Peru measures the efficiency of its lamps in watts lost. $E_I = \frac{P}{P_r}$, where $\Phi$ = lumens, $P$ = watts, and $P_r = 0.2\Phi$. $P_r \leq 0.24 \sqrt{\Phi} + 0.0103\ \Phi$.

Colombian Standard NTC 6199 [101] promotes cross-ventilation in buildings and adopted the ANSI/ASHRAE 62.1 Standard [102] for minimum ventilation rates setting an average ventilation rate of 4.52 l/s/person (SD of 0.65) for indoor spaces. The Mexican Voluntary Standard NMX-AA-164-SCFI-2013 [103], which deals with minimum criteria and environmental requirements for buildings, specifies that all buildings must provide natural ventilation; however, it does not specify any airflow rates. Furthermore, it establishes that all buildings that are mechanically ventilated must provide a natural ventilation alternative. In Chile, the NCh1973.Of87 standard [104] claims that high levels of condensation are produced due to (1) high relative humidity and (2) low temperature in the dwelling's surface. The standard mentions that some of the causes are insufficient air change and admission of external air with high levels of relative humidity (RH). It provides that buildings must meet the requirement established in Equation (1).

$$N > \frac{0.83 \times m_v}{(H_{is} - H_e)V}, \tag{1}$$

where $N$ is the number or air changes, $m_v$ is mass of water vapour produced in one hour inside the building, $H_{is}$ is internal RH $H_e$ is the external RH, and $V$ is the volume of the space.

Overall, the standards within the reviewed countries are aware of the importance of including natural ventilation strategies in buildings, but other than the international standards, most of them do not go into airflow rate calculations.

### 3.2.5. Air Conditioning

The usage of air conditioning (AC) amongst the reviewed countries varies, as some places within these countries are located in cold zones (south of Argentina and Chile) or in elevated places (Andean regions, Central Mexican plateau) where air conditioning is not needed. It is, however, one of the three largest energy consumers in buildings in LATAM [67]. For example, it is estimated that 20% of Mexican dwellings (5.5 M) have air-conditioning systems with 53% of a "minisplit" type and 46% a "window type" [105]. This figure is expected to increase due to climate change and global warming, and hence the importance of well-set TSs and AC rules in general. Furthermore, it is estimated that implementing ERSs in LATAM could reduce nearly 11% of the current energy consumption of the region [58]. Panama's TSs are under development, whereas Guatemala and Peru lack them completely. All the TSs related to air conditioning, where applied, are mandatory.

The international standard ISO 5151 [106] has been adopted as a benchmark for measuring the cooling capacity and energy efficiency of air conditioners in Colombia [107], Costa Rica [108], and Chile [109]. The energy efficiency ratio (EER) and the seasonal energy efficiency ratio (SEER) are the most commonly used units to measure energy efficiency in air-conditioning systems. The energy efficiency ratio (EER) is a dimensionless unit, described as the ratio between the cooling capacity to the power at full load (both measured in watts). The seasonal energy efficiency ratio (SEER) is designed to measure the performance at partial load, meaning that it considers the variations of external temperature and the effect of the cooling load. The higher the EER/SEER, the more efficient the appliance. Table 5 presents the different efficiencies for air-conditioning systems in the reviewed countries. The average EER amongst the countries with standards to achieve a class A/1 rating is 4.78. This number increases due to the high EER required in Mexico, where its energy standards are being homologated to those in the United States and Canada as part of the North American Free Trade Agreement (NAFTA) [110].

**Table 5.** Comparison of air-conditioning standards in the region. We show the air conditioning (AC) rating systems of the reviewed countries. Some use labels in (A, G) whereas others within (1,5). Both are shown. EER stands for energy efficiency ratio, SEER stands for seasonal energy efficiency ratio, CEE and CC stand for cooling capacity, and IEE stands for energy efficiency index (Indice de Eficiencia Energética). We also show (right-hand side) the status (S) of whether a minimum energy performance standard (MEPS) and an energy rating system (E) exist in each country. "-" stands for "no standard".

| Country | S | Rating System | | | | | | | Technology | | | |
|---|---|---|---|---|---|---|---|---|---|---|---|---|
| | | A/1 | B/2 | C/3 | D/4 | E/5 | F | G | Split | Compact | Window | Divided |
| ARG | M | 320 < EER | 300 < EER ≤ 320 | 280 < EER ≤ 300 | 260 < EER ≤ 280 | 240 < EER ≤ 260 | 220 < EER ≤ 240 | 220 < EER | MEPS and E | MEPS and E | - | - |
| BRA | M | 323 < CEE | 302 < CEE ≤ 323 | 281 < CEE ≤ 302 | 260 < CEE ≤ 281 | - | - | - | MEPS and E | - | MEPS and E | - |
| CHI | M | 320 < IEE | 320 ≥ IEE > 300 | 300 ≥ IEE > 280 | 280 ≥ IEE > 260 | 260 ≥ IEE > 240 | 240 ≥ IEE > 220 | 220 ≥ IEE | E | E | E | - |
| COL | M | 375 ≤ EER | 350 ≤ EER < 375 | 325 ≤ EER < 350 | 300 ≤ EER < 325 | 275 ≤ EER < 300 | | - | E | E | E | - |
| MEX [10] | M | CC < 1758 9.7 EER | 1758 ≤ CC ≤ 2345 9.7 EER | 2345 ≤CC ≤ 4103 9.8 EER | 4103 ≤CC ≤ 5861 9.7 EER | 5861 ≤CC ≤ 10548 8.5 EER | - | - | MEPS and E | - | MEPS and E | MEPS and E |
| | V | | | | | | - | - | | | | |
| PER | M | SEER ≥ 560 | 510 ≤ SEER < 560 | 460 ≤ SEER < 510 | 410 ≤ SEER < 460 | 360 ≤ SEER < 410 | 310 ≤ SEER < 360 | SEER < 310 | - | - | - | - |
| CRC | - | - | - | - | - | - | - | - | MEPS and E | - | MEPS and E | MEPS and E |
| GUA | - | - | - | - | - | - | - | - | - | - | - | - |
| PAN | - | - | - | - | - | - | - | - | - | - | - | - |

[10] Mexico is in the process of homologation with the standards of the United States; hence, we make reference to the new standard.

## 4. Discussion

LATAM has recorded a rapid change in its urban structure and relatively fast, national-level economic growth in the past 20 years. The countries of the region have taken measures against the acknowledged targets set in the Paris Agreements to reduce Greenhouse gas emissions (GHGs) and identified the built environment as a key element to achieve them.

Although the revised policies arise from the same common point and go towards the same goal, we find, unsurprisingly, that the impacts of these differ depending on the country. It is well documented that the effectiveness of a GBR varies from country to country [111], since it depends directly on external factors such as: levels of corruption, existing infrastructure to implement the rule, the country's (or even region's) level of development, etc.

There are countries that are already on a path towards reducing their energy consumption in buildings, but not necessarily through GBRs. First, we find that Costa Rica's "National Decarbonisation Plan" is leading with 100% of new buildings (including commercial, residential, and public) to use energy created from renewable sources. However, this is obviously not a fallout of their GBRs but rather with a national strategy of 100% electricity generation from renewables. Similarly, Peru's economy is expected to grow 6.5% in the next 10 years along with their energy consumption. So far, their strategies resulted in 35,638 Mt $CO_2$ emissions saved from 2009 to 2018 through renewable generation [112].

In contrast, Mexico's strategy towards reducing 50% of its GHGs by 2050 as compared to 2000 is still a long way from being on track. Current GBRs have the potential of saving enough energy to power 4.4% (1.5 M) of their total household stock for one year. It is unknown whether they are on track to achieve these targets as we could not find any implemented follow-up mechanisms to evaluate the rules. The same problem is repeated in Argentina and its IRAM standards. Energy-related GBRs often fail as the compliance of these is not a common practice due to lack of expertise, therefore becoming voluntary rules in reality. Some of these problems can be tackled with more attractive incentives and appropriate technical training. Naturally, unless systemic changes occur with regard to corruption, simply adding more rules will do little to significantly change outcomes.

Energy labelling systems have a potentially promising future in the region. With outcomes such as the mentioned energy reduction in Mexico [67], they may be viable alternatives prior to enacting mandatory laws. The strategy is simple. Since income levels in developing countries are on average lower than in the rest of the world, people will look for ways to spend less. This means that a product that can significantly reduce energy bills may be more attractive than the others, especially at cost parity. There is the tangible case of Mexico, where labelling only on refrigerators, washing machines, and air conditioners caused a decrease of 9.6% compared to the projected national electricity demand [67]. However, growing populations, increased new build, and rising incomes can rapidly "take back" these gains.

Figure 7 shows the discrepancy that still exists between countries with respect to the variety of green policies adopted (for example, Mexico or Chile vs. Guatemala). However, this does not necessarily mean that the countries with a larger number of policies adopted will have had larger energy reductions in buildings. In fact, Figure 7 may suggest the opposite. For instance, Costa Rica does not have a wide variety of green policies, but yet they have managed to generate their energy at 100% renewable, meaning that simple policies could produce larger impacts.

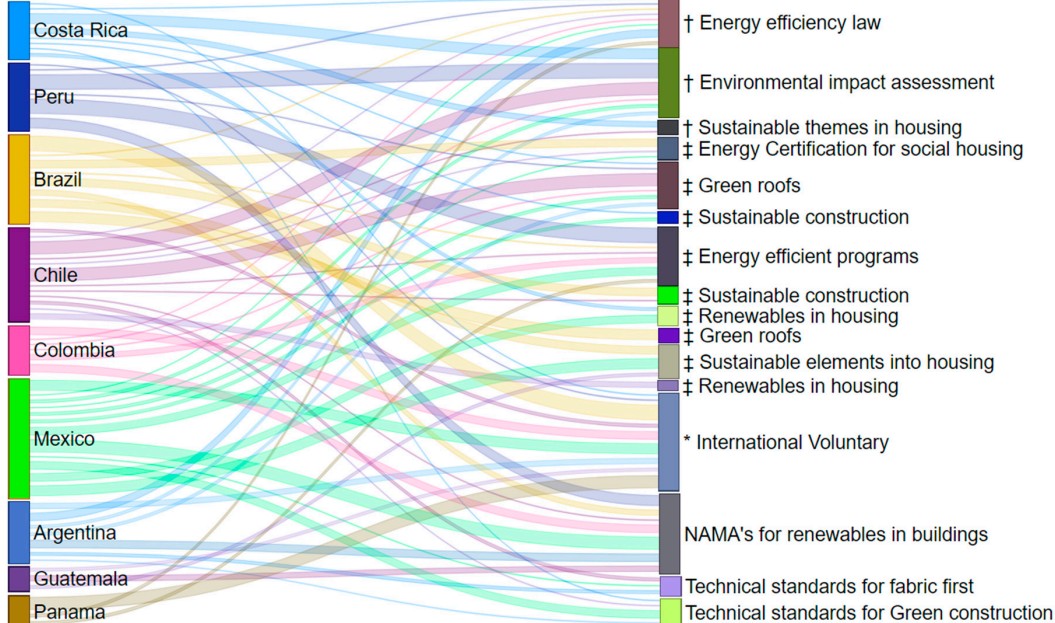

**Figure 7.** "Sankey" diagram that shows the situation of the green building regulations discussed in this article, (right side) and the corresponding country (left side) (legend = † command and control, ‡ economic incentives, * marketable permit system).

## 5. Conclusions

This paper evaluates the policy progress that countries in LATAM have made to curb build-related energy consumption and carbon emissions through the adoption or enactment of green building rules. The progress made suggests that climate change has become a national priority in many parts of LATAM. The adoption of such rules and continued increases in building rates, particularly through closing the housing deficit, could help "lock-in" the benefits of decarbonisation.

Given the uneven level of progress between countries, and the presence of shared language and cultures, our main conclusion is that there is great potential for the "cross-fertilisation" of GBRs between LATAM countries: Mexico's "green mortgage scheme" and Chile's PPEE strategy. The first has already been adopted in Colombia and has a great potential to spread across the region. In terms of building fabric, regulations must be tailored to the specific climate. This is only the case in Argentina and Chile, where the TSs of the Patagonian and Andean regions are specifically made against cold climates. Chile's strategy to ventilation could be looked at in detail by the other countries of the region as it is also the only one to regulate airtightness.

It is important for governments to not only implement the right rules and policies but also mechanisms to evaluate their success or failure to ensure a more sustainable future for the coming generations. Overall, we observe that while some progress has undoubtedly been made in the region, more holistic policies that consider whole building performance whilst maximising the potential for passive design are needed. We capture these in greater detail below. We, therefore, provide a set of recommendations to the policymakers of LATAM, drawn from the analysis of this paper.

- Technical standards should go beyond enacting. There is a need to ensure that there are enough qualified personnel to implement them so they are easily adopted. We, therefore, recommend that governmental offices must have enough human capital to adequately enforce the existing GBRs. Capacity building is essential through training; for example, adding future architects and engineers into these GBRs.
- EIs need to provide more attractive incentives to developers/builders and homeowners. We noticed that the common denominator amongst the policies with small/none positive outcomes were the ones with small unattractive incentives [49,54].

- EI-NAMA projects in Guatemala and Mexico have resulted in positive outcomes [57,63]. This may be due to the fact that policy-implementers are legally bound to provide follow-up reports. We, therefore, recommend stronger "follow-up" measures.
- We found that 20% of the found GBRs did not have any type of review or evaluation in place. We recommend that proper evaluations are undertaken for the GBRs currently lacking. These are shown in Appendix C of this document.
- We recommend adding airtightness standards to the current TSs from the region related to thermal comfort and heat loss in buildings to avoid unintended consequences such as poor indoor air quality.
- Most of the TSs found were adequate and well-designed. However, they were not found as part of the building code of the country. This means that even if, on paper, they are mandatory, the fulfilment of these standards is not a requirement to process a building permit at the local authorities. We strongly suggest an integration of these to their respective building codes so they become one of the requirements for a building permit.
- We encourage that all the technical standards must be open access to increase access, enable uptake, and breed a culture of transparency.
- Avoiding wholesale "importation" of GBRs to ensure they properly match local requirements, technical knowledge, and availability of skilled labour and materials has proved successful with the evolution of sustainable building standards in other parts of the world, such as India [113] and the Middle East [114].

**Author Contributions:** C.Z.-G. conducted the research collecting and investigating the reviewed rules, undertaking the analysis, and the reporting the results. S.N. supervised the overall research, articulated the core idea, designed the methods, suggested additions to key areas, and provided support in reviewing and drafting the paper. All authors have read and agreed to the published version of the manuscript.

**Funding:** This research was funded by the Mexican Council of Science and Technology (CONACyT) through its programme: 'Becas de postgrado al extranjero', and supported by the UK's Engineering and Physical Sciences Research Council (EPSRC) grant "Zero Peak Energy Building Design for India" (ZED-i, EP/R008612/1).

**Acknowledgments:** Carlos Zepeda-Gil would like to thank COMECYT (Consejo Mexiquense de Ciencia y Tecnología) for the support received under the program 'Beca de Apoyo Extraordinario, Folio: 16BAE0007'.

**Conflicts of Interest:** The authors declare no conflict of interest.

## Abbreviations

**Acronym List by Alphabetical Order**

| Acronym | Meaning |
| --- | --- |
| ASHRAE | American Society of Heating, Refrigerating and Air-Conditioning Engineers |
| ARG | Argentina |
| IRAM | Argentinian Institute of Standardisation and Certification |
| BRA | Brazil |
| BTU | British thermal unit |
| $CO_2$ | carbon dioxide |
| CHI | Chile |
| COL | Colombia |
| CAC | command and control |
| CC | cooling capacity |
| CSC | Costa Rica |
| EI | economic incentives |
| IEE | energy efficiency index |
| EER | energy efficiency ratio |
| ERS | energy rating system |
| EU | European Union |
| GWh | gigawatt hours |

## Abbreviations

**Acronym List by Alphabetical Order**

| Acronym | Meaning |
|---|---|
| GBR | green building rules |
| GUA | Guatemala |
| LATAM | Latin America |
| NOM | mandatory standardisation norm |
| MPS | marketable permit system |
| MEX | Mexico |
| M | million |
| Mt | million tonnes |
| MEPS | minimum energy performance standards |
| SEDATU | Ministry of Urban and Rural Development |
| CONAVI | National Housing Commission |
| PNCTE | National Programme of Greenhouse Gas Tradable Emission Quotas |
| NAMAs | nationally appropriate mitigation actions |
| NAFTA | North American Free Trade Agreement |
| PER | Peru |
| SEER | seasonal energy efficiency ratio |
| TS | technical standards |
| UNFCCC | United Nations Framework Convention on Climate Change |
| US | United States |
| USD | US dollars |

## Appendix A

Complete list of green building rules for the countries discussed in this paper. For the benefit of the Latin American readers, we retained the original names of the rules and numbers. However, we translated the description of the law for international audiences.

**Table A1.** List of the reviewed GBR's, listed by country's alphabetical order.

| Laws | Decrees | Standards | Description | Type of Policy |
|---|---|---|---|---|
| **9.1 Argentina** | | | | |
| **Law 13059/03** | - | - | Sets minimum thermal comfort requirements; | CAC |
| **Law 4428** | - | - | Promotes green roofs | IE |
| **Law 449** | - | - | Establishes impact evaluation studies prior to building | CAC |
| **Law 123** | - | - | Establishes the extents of the impact evaluation to be carried out | CAC |
| - | 1030/2010 | - | Establishes thermal comfort requirements to all types of buildings | CAC |
| - | 140/2007 | - | Declares energy efficiency as a subject of national priority | CAC |
| - | 222/2012 | - | Establishes impact evaluation studies prior to building | CAC |
| - | 140/2007-Anexo I, inciso 2.9 | - | Establishes the need for an energy rating system in buildings | CAC |
| - | - | IRAM 11900 | Result of Decree 140. Establishes methods for calculating energy efficiency in buildings | STANDARD |
| - | - | IRAM 11630 e IRAM 11659-1 | Standards for thermal and acoustic insulation in buildings | STANDARD |
| - | - | IRAM 11.603 | Climatic zoning | STANDARD |
| - | - | IRAM 11.605 | Allowances on thermal transmittance | STANDARD |

**Table A1.** *Cont.*

| Laws | Decrees | Standards | Description | Type of Policy |
|---|---|---|---|---|
| - | - | IRAM 11.625 | Risk assessment in condensation | STANDARD |
| - | - | IRAM 11659-2 | Thermal conditioning in buildings | STANDARD |
| - | - | IRAM 62404 | Lighting | STANDARD |
| - | - | IRAM 62406 | A/C | STANDARD |
| - | - | IRAM 210001-1 | Solar panels | STANDARD |
| - | - | IRAM 1739 | Insulating materials | STANDARD |
| - | - | IRAM 11.601 | Calculations on thermal transmittance | STANDARD |
| - | Argentina Carbon Tax—Impuesto al dioxido de carbono | - | - | MPS |
| **9.2 Brazil** | | | | |
| **Laws** | **Decrees** | **Standards** | **Description** | **Type of Policy** |
| Law 10.295 | - | - | Establishes maximum levels of energy consumption | CAC |
| - | No. 4.059 | No. 4.059 | Establishes energy rating systems in buildings | CAC |
| Law 8.666 | No. 7746/2012 | - | Green building strategies must be included in tenders for public buildings | CAC |
| - | No. 7.746 | - | Different criteria in energy use, water-saving, and low-environmental-impact materials | CAC |
| Lei Municipal (Local Law) 6.793/2010 | - | - | Reduces property tax by 20% over a 5-year period if the building adopted sustainable strategies | TAX |
| - | No. 3.5745/2012 | - | Promotes sustainable construction through fiscal incentives | IE |
| Qualiverde (Green Quality) certification | - | - | Fiscal incentives for having Qualiverde certification | IE |
| Tax over property purchase | - | - | Tax over property purchase is 4% of the property. However, it is reduced to 2% if the Qualiverde certification is obtained | TAX |
| Transformative Investments for Industrial Energy Efficiency (TI4E) | - | - | Creates energy efficiency projects on SMEs, including building appliances | IE-NAMA |
| **9.3 Colombia** | | | | |
| **Laws** | **Decrees** | **Standards** | **Description** | **Type of Policy** |
| Ley 145 | - | - | Establishes sustainable development as a priority issue from the national development plan | CAC |
| Ley 697/2001 (Law for the Promotion of Energy Efficiency and Renewable Energies) | Decree No. 3683 | - | Establishes energy efficiency as a national priority | IE |
| Law Project 119 | - | - | Grants fiscal incentives to green buildings | IE |
| Hipoteca Verde (Green Mortgage) | - | - | Grants loans for green technologies in social housing | IE |
| Local Agreement No. 186 | - | - | Proposes a green building code | IE |
| National Savings Fund Stamps | - | - | Green building rating system | IE |
| Programa Nacional de Cupos Transables de Emisión de Gases de Efecto Invernadero (Tradable Greenhouse Gas Emission Quotas) (PNCTE) | - | - | Carbon Pricing Scheme | MPS |

**Table A1.** *Cont.*

| Laws | Decrees | Standards | Description | Type of Policy |
|---|---|---|---|---|
| | | **9.4 Chile** | | |
| **Laws** | **Decrees** | **Standards** | **Description** | **Type of Policy** |
| **Ley 20.402** | - | - | Establishes the creation of the Chilean Agency of Energy Efficiency (AChEE) | CAC |
| - | No. 74 | - | Establishes the creation of a governmental interdisciplinary committee | CAC |
| - | Ley No. 458 | - | Includes aspects of sustainable urban planning | CAC |
| **Modification of Article 162** | - | - | Makes mandatory to include energy efficiency strategies in buildings | CAC |
| - | - | NCh2677:2002 | Promotes energy efficiency in public lighting | STANDARD |
| - | - | NCh3081:2007 | HVAC systems, rating, and labelling | STANDARD |
| - | - | NCh3082:2008 | Fluorescent lighting | STANDARD |
| - | - | NCh3149:2008 | Environmental design of buildings | STANDARD |
| - | - | Nch3184:2010 | Minimum standards and labelling of solar panels | STANDARD |
| - | - | NCh 3048/1:2007 | Establishes sustainability indicators in buildings | STANDARD |
| - | - | NCh3049/1:2007 | Promotes sustainability in building, methods, environmental performance, and materials | STANDARD |
| - | - | NCh3055:2007 | Guidelines for environmental quality | STANDARD |
| - | - | NCh3149:2008 | Energy efficiency in buildings | STANDARD |
| **Self-Supply Renewable Energy (SSRE)** | - | - | Fosters the usage of renewable energies in SMEs | IE-NAMA |
| | | **9.5 Mexico** | | |
| **Laws** | **Decrees** | **Standards** | **Description** | **Type of Policy** |
| **Federal Law of Climate Change** | - | - | Sets goals for the reduction of greenhouse gas emissions | IE, CAC, |
| **Law for Sustainable Energy Usage DOF 28-11-2008** | - | - | Promotes sustainable energy production and consumption | IE, CAC, |
| **Law of Housing DOF 23-06-2017** | - | - | Aims to reduce qualitative and quantitative housing deficit, and includes sustainable features | IE, CAC |
| - | - | NOM-007-ENER-2004 - | Promotes energy efficiency in public lighting | STANDARD |
| - | - | NOM-008-ENER-2001- | Building envelope and energy efficiency in nonresidential buildings | STANDARD |
| - | - | NOM-009-ENER-1995 | Promotes energy efficiency and thermal transmittance in industrial buildings | STANDARD |
| - | - | NOM-011-ENER-2006 | HVAC systems | STANDARD |
| - | - | NOM-018-ENER-2011 | Characteristics of insulation materials | STANDARD |
| - | - | NOM-020-ENER-2011 | Building envelope and energy efficiency in residential buildings | STANDARD |
| - | - | NOM-021-ENER/SCFI-2008 | Building envelope and energy efficiency in rooms with HVAC systems | STANDARD |
| - | - | NOM-023-ENER-2010 | Promotes energy efficiency in HVAC systems | STANDARD |
| - | - | NOM-24-ENER-2012 | Thermal characteristics of glass for building | STANDARD |
| - | - | NMX-AA-164-SCFI-2013 | Minimal criteria for sustainable buildings. Voluntary standard | STANDARD |

**Table A1.** *Cont.*

| Laws | Decrees | Standards | Description | Type of Policy |
|---|---|---|---|---|
| **Code for Sustainable Housing** | - | - | Economic incentives that suggest various sustainable strategies at the design and construction stage | IE-NAMA |
| **Energy Efficiency in SMEs as a Contribution to a Low-Carbon Economy** | - | - | Scheme that aims to reduce the energy consumption of all the aspects related to SMEs (including building) | IE-NAMA |
| **Articles 293 and 294 of the Financial Code of Mexico City** | - | - | Establishes fiscal incentives that promote sustainable construction | CAC |
| **Local Council of Zapopan** | - | - | Offers 100% discount on land property tax, to all LEED (Leadership in Energy & Environmental Design) projects | IE |
| **Secretary of Finance** | - | - | Offers fiscal incentives to companies that produce renewable energies | IE |
| **NADF-008-AMBT-2005** | - | - | Companies with more than 51 employees must have solar panels | CAC |
| **Ley del impuesto especial sobre producción y servicios (tax law over goods)** | - | - | Excise tax on the additional $CO_2$ emission content | MPS |
| **Mexico Pilot Emissions Trading System (ETS)** | - | - | Two-phase project to establish a fully developed ETS | MPS |
| **9.6 Peru** | | | | |
| Laws | Decrees | Standards | Description | Type of Policy |
| **Law No. 28245** | - | - | Establishes the creation of a National Strategy for Climate Change | CAC |
| **Law 28611** | - | - | General law of the environment. Establishes a right for a healthy environment | CAC |
| **Law 27345** | - | - | Declares efficient use of energy as a national priority | CAC |
| **Law 27867** | - | - | Establishes regional strategies for climate change and biological diversity conservation | CAC |
| **Law 26821** | - | - | Establishes targets for the sustainable use of natural resources | CAC |
| **Law No. 27345** | - | - | Aims to reduce the negative environmental impact of energy consumption | CAC |
| **Law 28611** | - | - | All constructions are subjected to environmental impact evaluation | CAC |
| **-** | Decreto Supremo No. 018-2006-VIVIENDA (housing) | - | Promotes sustainable urban development | CAC |
| **-** | National Policy of environment | - | Sets methods for achieving the goals set on Ley 28611 | CAC |
| **-** | Supreme Decree No. 053-2007-EM | - | Sets targets, methods, and evaluation for energy efficiency schemes | CAC |
| **-** | Supreme Decree No. 053-2007 | - | Promotes efficient energy use in residential buildings | IE |
| **-** | Supreme Decree No. 015-2012 | - | Establishes the right for housing | IE |
| **-** | - | D.S. No. 015-VIVIENDA(housing) | Guarantees right for green housing | TS |
| **-** | - | D.S. No. 003-2013-VIVIENDA (housing) | Handles efficient waste management during construction and demolition | TS |

**Table A1.** *Cont.*

| Laws | Decrees | Standards | Description | Type of Policy |
|---|---|---|---|---|
| | | **9.7 Guatemala** | | |
| Laws | Decrees | Standards | Description | Type of Policy |
| - | K´atun, Nuestra (our) Guatemala 2032 | - | Establishes the construction sector as a source of growth for the country | CAC |
| - | Decree No. 52-2003 | - | Promotes the development of renewable energy projects as well as fiscal incentives | CAC, IE |
| **Efficient Use of Fuel and Alternative Fuels in Indigenous and Rural Communities** | - | - | Provides financial and technical mechanisms to replace traditional appliances (i.e., firewood stoves) with "energy-efficient" ones | IE-NAMA |
| | | **9.8 Panama** | | |
| Laws | Decrees | Standards | Description | Type of Policy |
| **Law 69** | - | - | Promotes sustainable strategies from the design stage | CAC |
| **Política Nacional de Cambio Climático (National Policy for Climate change)** | - | - | Establishes general guidelines for climate change mitigation strategies | CAC, IE |
| | | **9.9 Costa Rica** | | |
| Laws | Decrees | Standards | Description | Type of Policy |
| **Law No. 7447** | - | - | Promotes rational use of energy | CAC |
| - | Decree No. 25584-96 | - | Establishes the minimum levels of energy efficiency | CAC |
| - | - | INTE/ISO 15392 | General principles in sustainable construction in buildings | STANDARD |
| - | - | INTE/ISO 21929 | Establishes indicators for an energy efficiency labelling system on buildings | STANDARD |
| - | - | INTE/ISO 21930 | Deals with construction materials | STANDARD |
| - | - | ISO/TR 21932 | Establishes definitions of sustainable construction | STANDARD |

## Appendix B

**Table A2.** Laws and decrees launched at the national level that induced the series of laws presented in this paper.

| Country | Policy | Year Promulgated |
|---|---|---|
| Argentina | Decree No. 140/2007 | 2007 |
| Brazil | National Strategy of Energy Efficiency | 2008 |
| Chile | National Strategy for Sustainable Construction | 2013 |
| Colombia | National Energy Plan 2050 | 2020 |
| Costa Rica | National Decarbonisation Plan | 2014 (Renewed in 2018.) |
| Guatemala | K'atun Development Plan, "Nuestra (our) Guatemala" 2032 | 2014 |
| Mexico | National Law for Climate Change | 2012 |
| Panama | National Strategy of Climate Change 2050 | 2019 |
| Peru | Referential Plan for the Efficient Use of Energy 2014–2025 | 2014 |

**Appendix C**

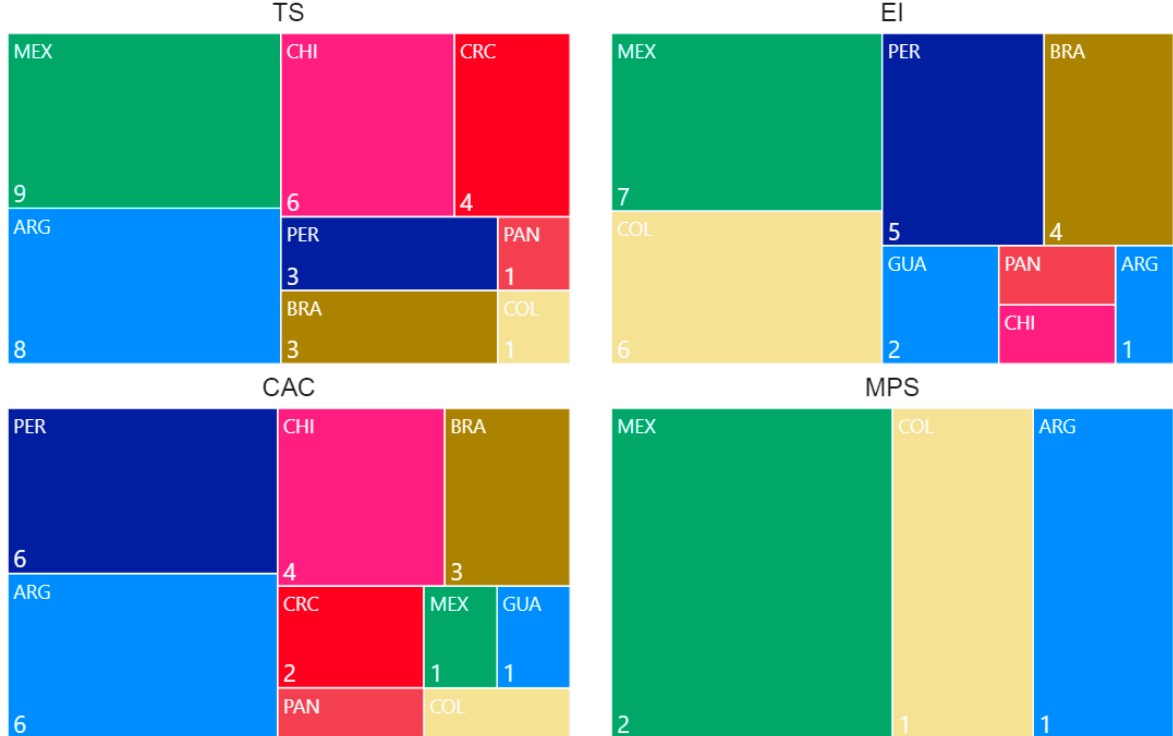

**Figure A1.** Map of the policies reviewed in this paper, framed according to our framework to classify green building regulations.

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
