# Peer review of "A Review of “Green Building” Regulations, Laws, and Standards in Latin America"

_buildings, doi:10.3390/buildings10100188_

Round 1
Reviewer 1 Report
Ms. Ref. No.: buildings-953416
Title: A review of “green building” regulations, laws, and Standards in Latin America
Overview and general recommendation:
National policies have an important role on mitigating the environmental impacts of the built environment. The manuscript overviews and analyzes the impacts and failures of those policies in South and Central America. The manuscript is well written and clear, despite some few typos and non-uniform use o IS of units, with a sounding systematic literature review. However, there are several issues that need to be corrected. For this reason, a major revision is recommended to be awarded. The comments in detail are listed below.
Comments:
1. In the Abstract, line 14, there is missing some sentence linking both sentences. “Hence” term presents a causation that seems illogical between two arguments. In other words, the growth in built environment is not the causation for this literature review.
2. In some instances, the manuscript has lump sum citations. Review each reference individually or remove redundant ones when possible.
3. Linkage to all Tables and Figures is missing, showing only an error message.
4. Not all units are according to the international system of units, both in their capitalization and terms.
5. On page 9, line 295, Figure 6 truncates the text.
6. Section 3 should also review heating, cooling, and renewable energy production in buildings. Air conditioning is presented as a subsection from Ventilation. This 3.2.3.2 subsection should focus primarily on mechanical ventilation.
7. In Discussion section, the implications of the findings and the limitations of the study must be presented and discussed.
8. The Conclusion section must present the main conclusions of the study before presenting recommendations to policy makers. In this form, it is not clear if a bullet point is a conclusion or a recommendation.
9. References must be revised to all references to be formatted according to the same reference style.
Author Response
We thank all reviewers for taking the time to review this paper and their useful comments. We note that the typographical error: “message: Error! Reference source not found.” seem to have occurred as a result of copying our word document to the template provided. We have double-checked in various computers for this problem and hope this does not persist with the present revision, and the reviewers have a more readable manuscript.
Please find our responses to specific comments below.
Reviewer One
Comment 1
- In the Abstract, line 14, there is missing some sentence linking both sentences. “Hence” term presents a causation that seems illogical between two arguments. In other words, the growth in built environment is not the causation for this literature review.
Response 1
Thanks for pointing this out. Now amended.
Comment 2
- In some instances, the manuscript has lump sum citations. Review each reference individually or remove redundant ones when possible.
Response 2
We had reviewed all the document in detail for “lump sum citations” and removed when these were not necessary, as seen below.
Original Text
31 the Global North to reduce energy consumption in buildings [5,6], as it is responsible for a third of the
42 As major emitters of carbon, buildings could play a key role in mitigating the effects of climate change [12,13],
43 from the construction stage [14], throughout their ‘useful life’ [15,16], to the end of their
289 These also provide a Cost Benefit Analysis with an amortization table that includes payback [54,55].
95(conclusions) such as India [93,94] and the Middle East [95].
Amended Text
31 the Global North to reduce energy consumption in buildings [5], as it is responsible for a third of the
42 As major emitters of carbon, buildings could play a key role in mitigating the effects of climate change [12],
43 from the construction stage [13], throughout their ‘useful life’ [14], to the end of their
289 These also provide a Cost Benefit Analysis with an amortization table that includes payback [54].
95(conclusions) such as India [93] and the Middle East [94].
Comment 3
- Linkage to all Tables and Figures is missing, showing only an error message.
Response 3
Amended.
Comment 4
- Not all units are according to the international system of units, both in their capitalization and terms.
Response 4
Amended, as seen below.
Original text
345 establish minimum levels of illumination levels in lux
374 minimum energy efficiency values for the reviewed countries in Lumen per watts (Lm/w) unless otherwise stated.
394 Standard for minimum ventilation rates setting an average of 4.52 L/s/person (st…
Amended text
345 establish minimum levels of illumination levels in lux (lx)
374 minimum energy efficiency values for the reviewed countries in Lumen per watts (lm/W) unless otherwise stated.
394 Standard for minimum ventilation rates setting an average of 4.52 l/s/person (st…
Comment 5
- On page 9, line 295, Figure 6 truncates the text.
Response 5
Amended
Comment6
- Section 3 should also review heating, cooling, and renewable energy production in buildings. Air conditioning is presented as a subsection from Ventilation. This 3.2.3.2 subsection should focus primarily on mechanical ventilation.
Response 6
There are no energy rating systems that focus on heating nor energy production. We added this statement to the main text.
Comment 7
- In Discussion section, the implications of the findings and the limitations of the study must be presented and discussed.
Response 7
We believe this has been done, as seen below.
Paragraph 1 and 2 of the Discussion:
Summarises the different factors that determine the effectiveness of a policy.
Paragraph 3
We describe a “success story” and emphasise their path, saying that it was more a national strategy, rather than green building rules.
Paragraph 4
We describe the case of some countries that had policies in place but are not meeting their set targets.
Paragraph 5
We describe an effective strategy that has brought positive outcomes.
Comment 8
- The Conclusion section must present the main conclusions of the study before presenting recommendations to policy makers. In this form, it is not clear if a bullet point is a conclusion or a recommendation.
Response 8
This has now been suitably amended.
Comment 9
- References must be revised to all references to be formatted according to the same reference style.
Response 9
We updated the reference style according to the MDPI Endnote style provided in the author notes of the journal.

Reviewer 2 Report
It is an interesting work, but several changes should be carried out:
1. The nine selected countries should be indicated in Abstract.
2. The authors should improve Introduction section, adding some articles from high impact journals.
3. There are too many subsections and sub-subsections.
4. The message “Error! Reference source not found.” appears several times in the text. The authors should revise that because the review paper was very difficult to read.
5. The authors should revise acronyms and add an acronym list. Moreover, it should be better not to use acronyms for countries in the text.
6. The laws, decrees and standards are in Spanish, Portuguese and English in the text. It should be better to write them in English in the text.
7. The authors should revise lines 294-295, the sentence is cut.
8. Is it necessary to write Eq. (1) in Section 3.2.3.1?
9. Section 5.1 should be integrated into Section 5.
10. The authors should avoid overkilling references. For instance, [18-21] or [36-39].
11. The authors should revise the units used. For instance, the units for EER in lines 404-405.
12. The authors should avoid using “we”, “our” or “us”.
13. The authors should revise numbers, using dots and commas properly.
14. The authors should revise subindexes.
15. The authors should avoid using footnotes.
16. Numbers with two decimals (or one decimal, at least) are more appropriate for the last column in Table 1.
17. The second Table 1 should be Table 3.
18. Figure 3 should be clarified or replaced with a table.
19. Figure 8 and Table 6 could be in the text, out of Appendices B and C.
20. The authors should improve the writing.
Author Response
Reviewer two
Comment 1
- The nine selected countries should be indicated in Abstract.
Response 1
Unfortunately, we are limited by the allowed length of the abstract. We have however moved them to earlier in the paper (they are now on the third paragraph, within the main introduction).
Comment 2
- The authors should improve Introduction section, adding some articles from high impact journals.
Response 2
Journals within the first 15 citations (covered in the introduction) include: Energy and Buildings, Proceedings of the National Academy of Sciences, and Energy Policy. These have impact factors of 5, 9.4 and 5.4 respectively.
Comment 3
- There are too many subsections and sub-subsections.
Response 3
Subsections have been removed
Comment 4
- The message “Error! Reference source not found.” appears several times in the text. The authors should revise that because the review paper was very difficult to read.
Response 4
Apologies. Amended.
Comment 5
- The authors should revise acronyms and add an acronym list. Moreover, it should be better not to use acronyms for countries in the text.
Response 5
Amended. We have now added an acronym list.
Comment 6
- The laws, decrees and standards are in Spanish, Portuguese and English in the text. It should be better to write them in English in the text.
Response 6
Amended. They now have been translated to English. However, we have retained the original name/numbering system to assist those in Latin American countries in being able to quickly identify the relevant law.
Comment 7
- The authors should revise lines 294-295, the sentence is cut.
Response 7
We could not find a cut sentence in the mentioned lines. Perhaps it was an artefact of pasting into the journal’s word template? We hope it is clearer this time.
Comment 8
- Is it necessary to write Eq. (1) in Section 3.2.3.1?
Response 8
One of the objectives of the paper is to create “cross-fertilisation” of successful policies within the region (see conclusion). It was interesting to note this equation as Chile was the only country with a particular strategy to deal with Natural ventilation and included it so the other countries could make use of.
Comment 9
- Section 5.1 should be integrated into Section 5.
Response 9
Amended.
Comment 10
- The authors should avoid overkilling references. For instance, [18-21] or [36-39].
Response 10
We removed “citation overkill” as much as possible. Please see Response 2 to reviewer 1.
Comment 11
- The authors should revise the units used. For instance, the units for EER in lines 404-405.
Response 11
The EER is a dimensionless ratio and has been defined as such. All other units have now been revised, amended and written according to the international system of units, as seen below.
Original text
345 establish minimum levels of illumination levels in lux
374 minimum energy efficiency values for the reviewed countries in Lumen per watts (Lm/w) unless otherwise stated.
394 Standard for minimum ventilation rates setting an average of 4.52 L/s/person (st…
Amended text
345 establish minimum levels of illumination levels in lux (lx)
374 minimum energy efficiency values for the reviewed countries in Lumen per watts (lm/W) unless otherwise stated.
394 Standard for minimum ventilation rates setting an average of 4.52 l/s/person (st…
Comment 12
- The authors should avoid using “we”, “our” or “us”
Response 12
It is now widely accepted to write academic papers in “Active voice”. Amongst its advantages, it brings readability, and the agent of action preceding the verb is clear and direct. We show below a statement from the Journal “Nature” that supports this.
“Nature journals prefer authors to write in the active voice (“we performed the experiment . . .”) as experience has shown that readers find concepts and results to be conveyed more clearly if written directly.”
Comment 13
- The authors should revise numbers, using dots and commas properly.
Response 13
Amended
Comment 14
- The authors should revise subindexes.
Response 14
Some subindexes have been now removed
Comment 15
- The authors should avoid using footnotes.
Response 15
Footnotes are usually incorporated into the main text by the journal office during proofing. We simply use it to keep the flow of the main text.
Comment 16
- Numbers with two decimals (or one decimal, at least) are more appropriate for the last column in Table 1.
Response 16
Amended
Comment 17
- The second Table 1 should be Table 3.
Response 17
Amended
Comment 18
- Figure 3 should be clarified or replaced with a table.
Response 18
Figure 3 is clarified from lines 133 to 137
Comment 19
- Figure 8 and Table 6 could be in the text, out of Appendices B and C.
Response 19
These were originally on the text. However, we realised that Table 6, due to its length, reduced readability and flow, so we decided to move it to the end as an appendix. We consider that Figure 8 fits better as an appendix, using our own map of policy evaluation.
Comment 20
- The authors should improve the writing.
Response 20
We would had been grateful if the reviewer would had provided with specific cases where the use of English language was not appropriate. We have carefully proof-read the paper again and made amendments wherever possible.

Reviewer 3 Report
The article shows interesting insight on the current situation in Central and Southern America.
Please avoid lumping references such as lines 41 and 54. Please mention each source for its novel contribution. Some of the mentioned ones can be avoided without affecting the study.
Check and consider what are the similarities and differences with other studies in the field, see the articles https://www.sciencedirect.com/science/article/pii/S1364032117313643
https://www.sciencedirect.com/science/article/pii/S136403211830501X
and others.
Enrich the conclusion highlighting what is possible to copy from a Country to another one within LATAM to be suggested to the policy makers.
Author Response
Reviewer three
Comment 1
- Please avoid lumping references such as lines 41 and 54. Please mention each source for its novel contribution. Some of the mentioned ones can be avoided without affecting the study.
Response 1
We had reviewed all the document in detail to remove as much “lump citations as possible”. Please see Response 2 to reviewer 1.
Comment 2
Check and consider what are the similarities and differences with other studies in the field, see the articles https://www.sciencedirect.com/science/article/pii/S1364032117313643
https://www.sciencedirect.com/science/article/pii/S136403211830501X
and others.
Response 2
Thanks for pointing these out. We now include references to both these and draw out the similarities and differences of our own work. This is presented as shown below:
lines 42-50 We point the differences between these studies and ours
lines 318-320 When we specifically deal with Energy rating systems.
Comment 3
Enrich the conclusion highlighting what is possible to copy from a Country to another one within LATAM to be suggested to the policy makers.
Response 3
Amended. We added a paragraph in the conclusions talking about the policies with the highest ‘cross-fertilization’ potential.

Round 2
Reviewer 1 Report
The authors have addressed my comments adequately and improved the overall manuscript. Therefore, I recommend to be awarded its publication.
Author Response
Comment 1
The authors have addressed my comments adequately and improved the overall manuscript. Therefore, I recommend to be awarded its publication.
Response 1
We thank you for taking the time in reviewing our paper, and for your very useful advice.
Reviewer 2 Report
The manuscript is very interesting and has been improved. The authors have addressed most of the suggested changes, but some changes are still needed:
1. The authors were asked to improve Introduction section. Some additional articles from high impact journals published in the last five years could be added.
2. Fourth-level headings should be avoided.
3. The message “Error! Reference source not found.” appears in the text. For instance, in lines 202 or 335.
4. The authors should revise acronyms. For instance, Peru is not defined in lines 52-53 and some of that countries are redefined in lines 109-111.
5. In lines 433-435, it says: “The EER is the ratio between the cooling capacity (in Btu units per hour) to the energy consumed measured at full load (in watts).” Therefore, EER is in Btu/h/W. If EER is a dimensionless ratio, the authors should indicate the two parameters in the same units.
6. Numbers with two decimals (or one decimal, at least) are more appropriate for the last column (column called “%”) in Table 1. Column called “Total” was right in the previous version.
7. When it was said that Figure 3 should be clarified or replaced with a table, the previous Figure 3 is the current Figure 5. Therefore, the authors were asked to clarify Figure 5.
8. The authors should improve the writing and revise several mistakes. For instance, “ours widens”, “35 638” or “that there exists”.
Author Response
Thank you for taking the time in reviewing our manuscript. We hope to have addressed your comments adequately.
Comment 1
- The authors were asked to improve Introduction section. Some additional articles from high impact journals published in the last five years could be added.
Response 1
We now added in the Introduction various references from high Impact Factor (IF) journals such as: Environmental Science & Policy (IF 4.76), Renewable and Sustainable Energy Reviews (IF 12.11), and Energy and Buildings (IF 4.86),
Comment 2
- Fourth-level headings should be avoided.
Response 2
Amended. Fourth-level headings (i.e., 4.3.2.1, or 1.2.3.4) have now been removed. We show ‘third level’ headings only when necessary.
Comment 3
- The message “Error! Reference source not found.” appears in the text. For instance, in lines 202 or 335.
Response 3
Thank you for noticing these typos. We had now removed these and checked for additional in detail.
Comment 4
- The authors should revise acronyms. For instance, Peru is not defined in lines 52-53 and some of that countries are redefined in lines 109-111
Response 4
Thank you for noticing this error. This has now been amended.
Comment 5
- In lines 433-435, it says: “The EER is the ratio between the cooling capacity (in Btu units per hour) to the energy consumed measured at full load (in watts).” Therefore, EER is in Btu/h/W. If EER is a dimensionless ratio, the authors should indicate the two parameters in the same units.
Response 5
Thank you for pointing this out. We have now simplified the language and made use of Watts.
Comment 6
- Numbers with two decimals (or one decimal, at least) are more appropriate for the last column (column called “%”) in Table 1. Column called “Total” was right in the previous version.
Response 6
We now had changed the column “Total” as it was in the previous version. The column called “%” Refers to percentages. We believe that it would be clearer to the reader if this was rounded to the nearest integer.
Comment 7
- When it was said that Figure 3 should be clarified or replaced with a table, the previous Figure 3 is the current Figure 5. Therefore, the authors were asked to clarify Figure 5.
Response 7
We have now added some text at at the start of Section 3.1 to better explain the figure. We think this figure is better than a table as it visually indicates the level of activity across the region, without sacrificing data transparency.
Comment 8
- The authors should improve the writing and revise several mistakes. For instance, “ours widens”, “35 638” or “that there exists”.
Response 8
Thank you for pointing these typos. They have been amended as seen below.
Original text
“however ours widens as it includes”
Amended text
“however ours is broader, as it includes”
Original text
“35 638
Amended text
“35,638”
Original text
“conclusion is that there exists great potential for “cross-fertilisation” “
Amended text
“conclusion is that there is a great potential for “cross-fertilisation” “
Reviewer 3 Report
The authors improved their manuscript making it suitable for publication.
Author Response
Comment 1
The authors improved their manuscript making it suitable for publication.
Response 1
We thank you for your useful advices, and for taking the time in reviewing this manuscript.